# A UNIQUE M-PATTERN FOR MICRO-EXPRESSION SPOTTING IN LONG VIDEOS

**Jinxuan Wang**[1]**, Shiting Xu**[1,2]**, Tong Zhang**[1,2*]
[1]South China University of Technology & Engineering Research Center of the Ministry of Education on Health Intelligent Perception and Paralleled Digital-Human
[2]Pazhou Lab, Guangzhou, China
{202221043813,csxst}@mail.scut.edu.cn, tony@scut.edu.cn

## ABSTRACT

Micro-expression spotting (MES) is challenging since the small magnitude of micro-expression (ME) makes them susceptible to global movements like head rotation. However, the unique movement pattern and inherent characteristics of ME allow them to be distinguished from other movements. Existing MES methods based on fixed reference frame degrade optical flow accuracy and are overly dependent on facial alignment. In this paper, we propose a skip-$k$-frame block-wise main directional mean optical flow (MDMO) feature for MES based on unfixed reference frame. By employing skip-$k$-frame strategy, we substantiate the existence of a distinct and exclusive movement pattern in ME, called M-pattern due to its feature curve resembling the letter 'M'. Based on M-pattern and characteristics of ME, we then provide a novel spotting rules to precisely locate ME intervals. Block-wise MDMO feature is capable of removing global movements without compromising complete ME movements in the early feature extraction stage. Besides, A novel pixelmatch-based facial alignment algorithm with dynamic update of reference frame is proposed to better align facial images and reduce jitter between frames. Experimental results on CAS(ME)$^2$, SAMM-LV and CASME II validate the proposed methods are superior to the state-of-the-art methods.

## 1 INTRODUCTION

Spontaneous micro-expression (ME) is unintentionally leaked when people attempt to conceal or restrain their expressions in an intensely emotional situation (Li et al., 2023a). These expressions are of low intensity and usually last less than 0.5 second (Ben et al., 2022), making them difficult to be detected. Unlike macro-expression (MaE), once ME appears, it can reflect the most genuine emotion, which is why ME plays an indispensable role in the fields of psychology, business negotiation, and criminal investigation (Xu et al., 2022). Micro-expression spotting (MES) refers to locate segments (i.e., the onset and offset frame) of micro expressions in a long video. Global movements, including head movements, zooming in and out are almost inevitable and numerous over a extended time span. The brief and subtle nature of ME makes it almost imperceptible to the naked eye. Under such circumstances, ME is easily eclipsed by prolonged and large global movements.

With the advancement of computer vision (Zhang et al., 2024; Xu et al., 2023; Zhang et al., 2023), MES research has evolved mainly into traditional and deep learning methods. The latter primarily draws inspiration from the fields of temporal action localization (Yang et al., 2020), object detection (Tan et al., 2020) and visual recognition (Wang et al., 2021; Liu et al., 2024). Considering the enormous variety of expression scales, feature pyramid network (FPN) is widely employed in MES (He et al., 2022; Yu et al., 2021). However, deep learning methods are limited in various aspects, e.g., inadequate ME samples, disturbance from global movements and limitations of unimodal data. Recent developments in self-supervision and generative modeling can mitigate these flaws. Traditional methods resort to a prior knowledge of the ME field (Ekman & Friesen, 2019; Bhushan, 2015) and mainly include three steps: (1) facial alignment, (2) feature extraction, and (3) spotting. Facial alignment is essential (Yap et al., 2022) as it aims to remove head movements. Mainstream optical

---
*Corresponding Author

flow (OF) based alignment methods (Yuhong, 2021; Zhao et al., 2022) align all frames with a fixed reference frame (RF). However, due to massive global movements, OF cannot characterize subtle displacements properly. Dominate feature extraction approaches similarly based on fixed RF, which compute the OF between each frame and the RF to describe the changes between them (Yuhong, 2021; Zhang et al., 2020). These methods overly rely on the performance of facial alignment, as head posture and facial appearance differences between subsequent frames and RF are too significant to satisfy OF assumptions. Another problem is that these methods prefer to extract feature within the regions of interest(RoIs) to capture subtle movements selectively and reduce computation time, causing the inability to observe global movements. Thus many false micro movements are wrongly detected as ME. Spotting is designed to locate ME intervals. Zhang et al. (2020) utilize the property that ME moves in opposite directions during the occurrence and restoration phases to reduce false positives (FP). Li et al. (2023b) exploit the localization character of ME for spotting. But they all fail to fully utilize the characteristics of ME to distinguish them from other micro movements.

Based on the limitations above, we adopt an unfixed RF strategy for both facial alignment and feature extraction. We first propose a pixelmatch-based facial alignment algorithm with dynamic update of reference frame to improve alignment performance and reduce jitter. In order to minimize the disturbance of global movements, we propose a skip-$k$-frame block-wise main directional mean optical flow (MDMO) feature based on unfixed RF. Block-wise MDMO is capable of simultaneously extracting features and eliminating global movements. By using a skip-$k$-frame strategy, we substantiate a unique movement pattern of ME, called M-pattern. M-pattern is capable of precisely describing the complete ME movement from occurrence to restoration, and it is targeted and accurate for spotting ME intervals even without facial alignment. Then we provide a novel spotting rules based on M-pattern and three characteristics of ME to spot ME intervals.

## 2 RELATED WORKS

Facial alignment is widely used for its significance in eliminating global movements. Yan et al. (2014) utilize Local Weighted Mean(LWM) transformation to align the facial landmarks of each frame with a manually selected RF, but this can cause severe facial deformation. Yap et al. (2020) apply the online toolbox OpenFace (Baltrusaitis et al., 2018) for facial alignment. Zhao et al. (2022) and He et al. (2020) utilize OF as a displacement measure, and align each frame with the RF (normally the first frame) according to the displacement in the nose region. The above methods all adopt a fixed RF strategy, with better alignment for nearby frames but worse for distant ones.

Common feature descriptors include MDMO (Liu et al., 2016), LBP-TOP (Tran et al., 2019), HOOF (Verburg & Menkovski, 2019), etc. Main directional mean optical flow (MDMO) (Liu et al., 2016) is one of the most representative features in MES. It divides the OF field into 8 equally sized and disjoint bins and averages the OF vectors in main direction, hence obtains a more robust and representative feature to describe movement. Zhang et al. (2020) divide the OF field into 4 unequally sized and disjoint bins, which is more in line with the directions of ME movement. ME reveals itself in specific facial regions (Ma et al., 2019), so it is redundant to analyze the whole face. Existing methods mainly focus on features around eyebrows and mouth. Yuhong (2021) extracts 14 RoIs to analyze MDMO in each RoI. Zhang et al. (2020) extract feature in 12 RoIs to generate a 3D matrix to analyze spatio-temporal information. However, these localized extraction methods prevents them from observing the global movement, leading to many of them being recognized as ME.

Two solutions are available for addressing the disadvantages of fixed RF strategy: clip-based and unfixed-RF-based. Clip-based methods divide a whole video into multiple clips and perform MES on each clip individually. Wang et al. (2021) propose a clip proposal network to filter negative samples. Yang et al. (2021) and Tran et al. (2021) employ a sliding window strategy to scan across the video sequence. Li et al. (2023b) extract temporal feature over 300 ms (based on ME duration), and they discover a S-pattern in feature curve when ME occurs. However, the S-pattern can only describe half of the ME movement from onset to apex. Unfixed-RF-based methods implement a fixed interval to calculate the difference between two frames. Han et al. (2018) and Li et al. (2018) compute the difference between current frame $f_i$ and the average feature frame (i.e., the average feature of $f_{i-k}$ and $f_{i+k}$) to spot apex frame. Liong et al. (2021) extract OF feature between $f_i$ and $f_{i+k}$ instead of $f_i$ and RF. Due to the short time span between $f_i$ and $f_{i+k}$, it ensures the OF can correctly depict the movement between them. However, they ignore the fact that during the ME restoration phase, computing OF in this way leads to an additional peak in feature curve.

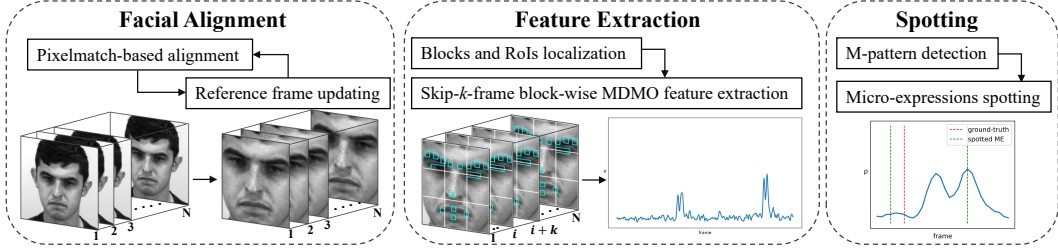

Figure 1: Overall framework. The proposed methods consist of three parts focusing on MES and global movements elimination. First, we introduce a pixelmatch-based facial alignment algorithm with dynamic update of reference frame to align facial images. Second, we propose skip-$k$-frame block-wise MDMO to extract features with subtle changes and eliminate global movements simultaneously. Under skip-$k$-frame strategy, we verify a unique movement pattern of ME, called M-pattern. Third, based on M-pattern and ME characteristics, we provide novel ME spotting rules.

## 3 PROPOSED METHOD

### 3.1 FACIAL ALIGNMENT

Aligning all frames with a fixed RF in long videos is not recommended and the OF-based method is significantly influenced by head displacement. Thus we propose a pixelmatch-based facial alignment algorithm with dynamic update of reference frame. Pixelmatch(Mapbox, 2022) is a Python package used for fast pixel-level image comparison. It is based on the RGB difference and returns the number of changed pixels between two images, denoted by $m$. As shown in Fig. 2, the positions of the matching and cutting box are defined on the basis of center coordinate $c(x, y)$.

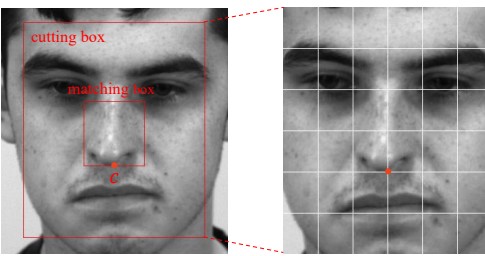

Figure 2: Cutting box and matching box.

The sizes of boxes are defined on the basis of two measures of the human face: horizontal distance between the inner eye corners and vertical distance between the nasal spine point and the line connecting the inner eye corners(Li et al., 2018). The nose is chosen as the matching box due to its stability and rigidity. The algorithm consists of alignment and reference frame update (pseudo-code and details in Appendix A.1). Alignment is the process of adjusting the center coordinate until $m$ between the current matching box and the reference matching box is minimized. $M_{min}$ is set to reduce the jitter between images, which means no adjustment will be made if the mismatch pixels is less than $M_{min}$. Since the head posture changes over time, it is necessary to adjust the RF dynamically. But we must ensure that the updated RF is well-aligned to prevent subsequent frames from aligning with a misaligned frame. If $m$ is equal to 0 or greater than $M_{max}$, the RF is updated as the current frame. $m = 0$ means perfectly aligned. $m > M_{max}$ implies a significant change in head pose or facial appearance, making subsequent frames difficult to align if the RF is fixed. If $0 < m \le M_{max}$, the RF will not be updated temporarily, until the frame with a more satisfactory alignment appears.

### 3.2 M-PATTERN

With a short time span, skip-$k$-frame strategy ($k = (N + 1)/2$, $N$ is the average ME duration) ensures that the two frames involved in computation satisfy the assumptions of OF (minor displacement and constant apparent brightness) even without facial alignment. We denote the $j$th RoI of the $i$th frame as $f_i^j$ and the OF magnitude between $f_{i-k}^j$ and $f_i^j$ as $\rho_i^j$. The calculation details will be discussed in Section 3.3. As shown in Fig. 3, we define a complete ME movement from $f_i$ to $f_{i+2k}$, with $f_{i+k}$ being the apex frame. Theoretically, two neutral states $f_i$ and $f_{i+2k}$ are identical with lower magnitude (i.e., lower muscle deformation intensity). The intensity to increase monotonically as the expression develop from a neutral state ($f_i$) to a peak state ($f_{i+k}$). Then it decreases as the expression restores from apex back to another neutral state($f_{i+2k}$). Under skip-$k$-frame ap-

proach, the feature curve of such ME exhibits a particular pattern, called M-pattern, as shown in Fig. 4. The details(❶∼❺) are described below.

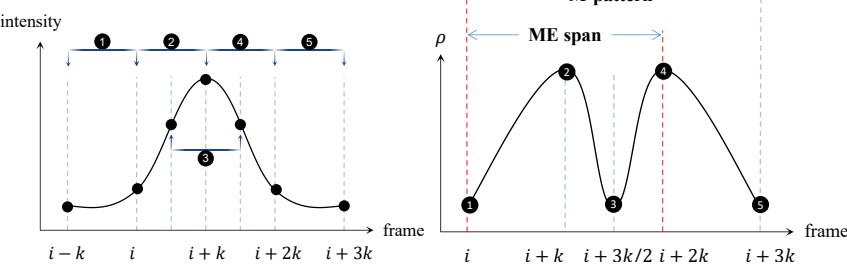

Figure 3: The muscle deformation intensity curve of a complete ME.

Figure 4: Skip-$k$-frame OF magnitude of a complete ME.

Point ❶ denotes the OF magnitude between $f_{i-k}$ and $f_i$. The expression is in the initial stage of occurrence and the facial state is relatively stable before $f_i$. Therefore $\rho$ is nearly 0.

Point ❷ is the first peak in feature curve. $\rho$ begins to increase as the ME develops from $f_i$ to $f_{i+k}$, where $f_i$ is in a neutral appearance, and $f_{i+k}$ is in the highest intensity of the facial motion.

Point ❸ is the local minimum in this feature curve. The onset-apex and apex-offset phases of ME are actually opposite and symmetrical movements. Therefore, $\rho$ begins to fall until it reaches a local minimum value in the vicinity of $f_{i+3k/2}$. For example, consider a frowning expression, where the halfway-furrowed appearance in lowering eyebrows phase closely resembles the halfway-furrowed appearance in raising eyebrows phase.

Point ❹ is the second peak in feature curve. $\rho$ starts to increase again because one of the frames is progressing towards the apex($f_{i+k/2} \rightarrow f_{i+k}$), while the other is recovering towards neutral ($f_{i+3k/2} \rightarrow f_{i+2k}$). In fact, ❶∼❸ and ❸∼❺ are y-axis-symmetric as the ME curve in Fig. 3 is also y-axis-symmetric.

Point ❺ is almost analogous to ❶. The facial appearance has fully restored to a neutral state, thus little difference can be found between $f_{i+2k}$ and $f_{i+3k}$.

Eventually, we get a trajectory shaped like 'M', with a span from $f_i$ to $f_{i+3k}$. But notice that although we get a feature curve with two waves, it is obtained by one single ME, and the final predicted interval is the start of the first wave to the peak of the second wave, not the entire 'M'.

CASME II(Yan et al., 2014), CAS(ME)$^2$(Qu et al., 2018) and SAMM-LV(Yap et al., 2020) are applied to the validation of the M-pattern. Appendix A.2 illustrates the feature curves of ME on three datasets. We can find that they all display the same pattern similar to letter 'M'. Thus we can utilize the M-pattern for MES.

### 3.3 SPOTTING RULES

We aim to exploit the unique properties of M-pattern to differentiate ME from other movements. Thus, we propose spotting rules based on M-pattern and ME characteristics(Subtlety, Localization, Transience). It is divided into 6 steps specifically.

Step 1. Peak detection and interval generation. These are applied to locate the peaks with potential rapid facial movements. The threshold is calculated as:

$$T = C_{mean} + p \times C_{std}, \tag{1}$$

where $C_{mean}$ and $C_{std}$ are the average and standard deviation of all local maxima in a video. $p$ is a parameter to determine the threshold level. The set $C^T$ comprises local maxima greater than $T$. As shown in Fig. 5, For each peak in $C^T$, we extend the left and right of $p_i$ to the relatively flat part of feature curve(or extend to trough like the middle of 'M', if there is an adjacent wave next to $p_i$), to generate an interval $w_i = [w_i^s, w_i^e]$.

Step 2. Potential ME spotting based on M-pattern. According to the inference of M-pattern, if two intervals (waves) are close enough to each other, i.e., satisfying Eq. (2):

$$w_{i+1}^s - k/2 < w_i^e, \tag{2}$$

we can combine them into one interval from the start of the first wave to the peak of the second wave. This is the combination rule when the ME is at an average length. Practically, the duration of some MEs is too short/long, and the duration of apex-offset phase of most MEs is longer than the onset-apex phase (Yan et al., 2013). In these cases the combination rules will be slightly different. For example, if a ME is from $f_i$ to $f_{i+k}$, the predicted interval of this ME is from the start to the end of the first wave since the second wave of 'M' is actually a replica of the first one. We use $l_{a,b}$ to denote the distance from a to b. The specific combination rules can be determined by the lengths of two adjacent waves:

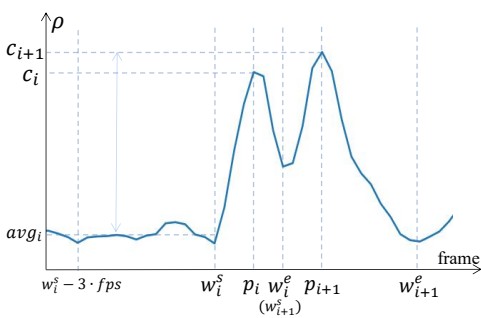

Figure 5: Symbolic markers of feature curve.

$$\begin{cases} [w_i^s, w_i^e], & \text{if } l_{w_i^s, w_i^e} \leq k, \\ [w_i^s, w_i^s + 2l_{w_i^s, p_i}], & \text{if } k < l_{w_i^s, w_i^e} \leq 2k, \ l_{w_{is}, p_i} \leq k, \\ [w_i^s, p_{i+1}], & \text{if } k < l_{w_i^s, w_i^e} \leq 2k, \ l_{w_i^s, p_i} > k, \\ [w_i^s, p_{i+1}], & \text{if } l_{w_i^s, w_i^e} > 2k. \end{cases} \tag{3}$$

Based on M-pattern, a complete ME consists of two waves, thus $w_{i+2}$ is the next to be analyzed after $w_i$, not $w_{i+1}$. In fact, not all MEs have a restoration phase. There is a circumstance when the expression reaches the peak state and then continues to remain in that state until the next expression occurs. The latter half of the curve in Fig. 3 will remain at the peak level. The feature curve of this type of ME has only one wave, instead of two waves like the M-pattern. Thus, if $w_i$ does not satisfy Eq. (2), we determine whether $w_i$ is a single wave according to Eq. (4).

$$w_{(i-1)}^e + k < w_i^s \text{ and } w_{(i+1)}^s - k > w_i^e. \tag{4}$$

In this case, it is easy to deduce the predicted interval is the ascending part of the wave, i.e., $[w_i^s, p_i]$. And $w_{i+1}$ is the next to be analyzed, not $w_{i+2}$.

Step 3. First selection based on subtle and localized nature of ME. $E^j = \{e_i^j | i = 0, 1, ...\}$ denotes the set of ME intervals of $j$th RoI after step 2, and $c_i^j$ is the peak value of each ME interval. If $e_i^j$ is generated by a double wave, $c_i^j$ is the larger one of two peaks. Subtlety means that the intensity of ME is much gentler than MaE and other movements. So we set a maximum threshold $T_{max}$ to exclude intervals with large $c_i^j$. Existing spotting rules treat each RoI individually, but the expression are the result of coordinated movements across multiple RoIs. For a MaE consisting of multiple RoIs, some of which have intensity greater than $T_{max}$ and some less than $T_{max}$, it is impossible to exclude such MaE if each RoI is treated separately. Thus, we reduce FP at the expression level, not at the RoI level. Specifically, intervals with IoU greater than 0.5 belong to the same movement. We introduce $ddeg$ to describe the overall facial muscle deformation degree for an expression, and $ddeg_i^j$ is the deformation degree for $j$th RoI, $i$th interval:

$$ddeg_i^j = c_i^j / avg_i^j, \tag{5}$$

where $avg_i^j$ is the average of $3 * fps$ frames preceding the start frame of $e_i^j$. $ddeg$ is the sum of $ddeg_i^j$ belonging to the same movement. If $ddeg > T_{ddeg}$ or the largest $ddeg_i^j$ in the same movement is greater than $T_{max}$, all $e_i^j$ belonging to this movement will be excluded simultaneously. The second feature is localization, which is reflected in the fact that ME involves fewer RoIs. If a movement engages more than half of the RoIs, then all $e_i^j$ constitute this movement are excluded.

Step 4. Second selection based on global movements. The movement of nose tip region is approximated as global movement because of its rigidity. Steps 1-2 are applied to the nose tip region to detect possible global movements, and $e_i^j$ will be excluded if it intersects with global movements.

Step 5. Merging. The merging involves fusing intersecting intervals, i.e., intervals belonging to the same expression, from different RoIs. Suppose there are two intervals $\boldsymbol{a} = [a^s, a^e]$ and $\boldsymbol{b} = [b^s, b^e]$ from different RoIs with IoU $iou_{\boldsymbol{ab}}$. If $iou_{\boldsymbol{ab}} > IoU_{merge}$, the two intervals are merged into one $[a, b]$, where

$$\begin{aligned} a &= max(a^s, b^s) - 0.5 \times (max(a^s, b^s) - min(a^s, b^s)), \\ b &= min(a^e, b^e) + 0.5 \times (max(a^e, b^e) - min(a^e, b^e)). \end{aligned} \tag{6}$$

This is an extension to the left and right with the intersecting intervals as the center. It ensures the length of merged interval is almost equal to the average length of intervals involved in merging.

Step 6. Last selection based on transient nature of ME. Transience refers to the short duration of the ME ($<0.5$s) (Yan et al., 2013), which allows us to exclude intervals with duration longer than 0.5s.

Hence, we have obtained all the predicted expression intervals that both satisfy the M-pattern and all the characteristics of ME.

### 3.4 FEATURE EXTRACTION

Since global movements are unidirectional, i.e., no restoration phase, the feature curves will be single-wave, the same as the feature curve of ME without a restoration phase. Thus, we propose the block-wise MDMO to extract features within RoIs and eliminate global movements in the early feature extraction stage.

Specifically, as shown in Fig. 6(a), we divide the facial area into 9 blocks to analyze the global movement and extract 15 RoIs, including 2 eye regions to detect ME movements and blink. Original MDMO(Liu et al., 2016) divides the OF field into 8 disjoint bins on average in Fig. 7. However, if the direction of movement falls between two adjacent bins, MDMO will only average the vectors within the bin with maximum count(i.e., the main direction), resulting in deviation in both magnitude and orientation of MDMO.

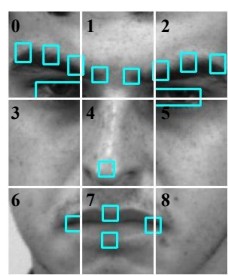

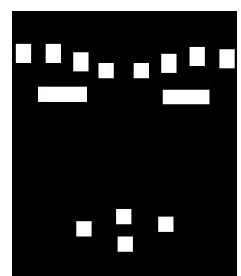

(a) Blocks and RoIs division.

(b) Mask.

For this reason, we divide the OF field into 8 intersecting bins as illustrated in Fig. 6(c). Each pixel will belong to two different bins, and directions that were between two adjacent bins can now fall into one bin. Each bin in conjunction with its two intersecting bins constitutes a direction group. For each block $R^k(k = 0, 1, ..., 8)$, we classify all the OF vectors into 8 bins according to their orientation and compute MDMO. Denote the pixel counts distribution in $R^k$ as $B^k = (b_0^k, b_1^k, ..., b_7^k)$. $\theta^k = argmax(b_0^k, b_1^k, ..., b_7^k)$ is the main direction. The MDMO of $R^k$ is compute as:

$$\overline{\boldsymbol{u}}^k = \frac{1}{|B_{\max}^k|} \sum_{\boldsymbol{u}^k(p) \in B_{\max}^k} \boldsymbol{u}^k(p), \quad (7)$$

where $|B_{max}^k| = max(b_0^k, b_1^k, ..., b_7^k)$, and $B_{max}^k$ is the set of OF vectors in main direction. $B = (B^0, B^1, ..., B^8)$ and $\Theta = (\theta^0, \theta^1, ..., \theta^8)$ provide a very clear description of the global movement. Similarly, MDMOs of 15 RoIs, denoted as $\overline{\boldsymbol{u}}^j$, are computed following Eq. (7). Subsequently, it is common(Zhang et al., 2020) to remove the global movement using the displacement in nose tip

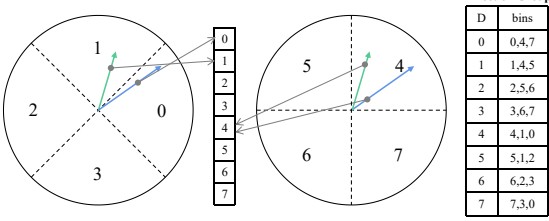

(c) 8 intersecting bins of optical flow field.

Figure 6: Block-wise MDMO.

region. However, The OF vectors distribution in Fig. 8 illustrates that this turns out to be limited to the translational head movements. For circumstances such as head rotation, brightness change, zooming in and out, etc., it can only lead to worse results. Therefore, we define five states (order, disorder, rotation, adjustment, blink) to categorize each frame according to $B$ and $\Theta$ with the purpose of eliminating global movements without disrupting the complete ME movements.

**Order**. Order means all $\theta^k$ point at the same direction group and 8 elements in $B^k$ for each block should be ordered and consistent with $\theta^k$, i.e., 3 bins in the direction group should contain all the OF vectors in $R^k$ and the remaining 5 bins should be close to 0. Normally, frames with translation and other minimal head movements will be classified as order. The MDMO of the nose tip region is basically in line with the global movement. Therefore we still subtract $\overline{\boldsymbol{u}}^{nose}$ to remove the global

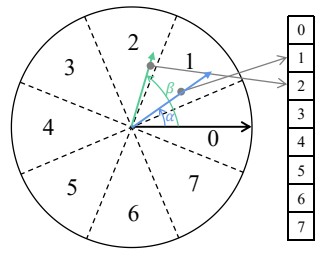
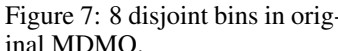

Figure 7: 8 disjoint bins in original MDMO.

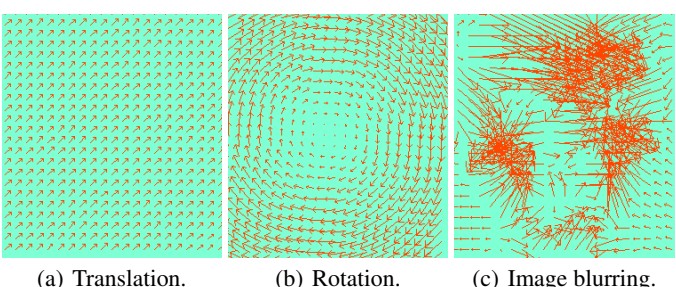

(a) Translation.      (b) Rotation.      (c) Image blurring.

Figure 8: Optical flow images of different circumstances.

movement. The final OF feature for $f_i^j$ is computed as:

$$\tilde{\boldsymbol{u}}_i^j = \overline{\boldsymbol{u}}_i^j - \overline{\boldsymbol{u}}_i^{nose}. \tag{8}$$

**Disorder**. Disorder means that the OF vectors are chaotic and random. Using $B$ alone is sufficient for determining whether a frame is disorder. Sometimes all $\theta^k$ points in the same direction group, but the OF vectors within each block are very disorganized. For $R^k$, if $min(B^k) > T_{disorder}$, then $R^k$ is disorder. In other words, each bin (direction) has a significant portion of vectors belonging to it. Brightness change and image blurring are most likely to cause disorder. In this case, MDMO of neither the blocks nor the nose tip can accurately represent their movement, but the MDMO of RoIs happen to have a moderate magnitude to generate a fake ME movement. Therefore, for all RoIs, we set their final OF as the average OF in a video:

$$\tilde{\boldsymbol{u}}_i^j = \frac{1}{N-k} \sum_{i=k+1}^{N} \overline{\boldsymbol{u}}_i^j. \tag{9}$$

**Rotation**. After facial alignment, the head rotation is almost centered on the nose tip. Excluding center block 4, the main directions of the other 8 blocks are mostly different. If we visualize 8 MDMOs with arrows, they should overall conform to a clockwise or counterclockwise circle or oval(See Fig. 8(b)). Head rotation is most likely to generate a fake ME movement as its magnitude is very close to ME. So the final OF feature is the same as Eq. (9).

**Adjustment**. When 9 $\theta^k$ do not all belong to the same direction group and the global movement is not rotation or disorder, simply following Eq. (8) doesn't really help. We use $\overline{\boldsymbol{u}}^k$ to represent the local movement of $R^k$, and $\overline{\boldsymbol{u}}^j$ should minus the MDMO of the block RoI $j$ belongs to, not $\overline{\boldsymbol{u}}_i^{nose}$, to eliminate the global movement.

$$\tilde{\boldsymbol{u}}_i^j = \overline{\boldsymbol{u}}_i^j - \overline{\boldsymbol{u}}_i^k, (f_i^j \in R_i^k). \tag{10}$$

**Blink**. Neutral blinking sometimes causes a slight vibration in eyebrows with other RoIs beyond the eyebrows being not affected. It is easy to detect blink using the left and right eye region since there are only upward and downward movements of the upper eyelid. If blink happens under an ordered state, the final OF of RoIs in eyebrow area should follow Eq. (9), and other RoIs should follow Eq. (8).

If frames within a ME span are classified as one of the other four states except for 'order', the M-pattern probably will be compromised since their OF vectors are altered. If there is a chance that $f_i$ is during a ME movement, the feature of this frame should be calculated according to Eq. (8), regardless of the state it is categorized into. The subtle nature of ME is reflected in the fact that muscle changes are largely confined to the RoIs, with no noticeable changes occurring in other facial region. This is why pixels within 15 RoIs are not involved in the calculation of the MDMO for 9 blocks, i.e., only the black area in Fig. 6(b) participates in the calculation of $\overline{\boldsymbol{u}}^k$. $\overline{\boldsymbol{u}}^k$ is used to describe the local movement of $k$-th block. Excluding pixels within the RoIs does not affect the final result since $\overline{\boldsymbol{u}}^k$ is averaged, but including pixels within the RoIs when a ME occurs makes $\overline{\boldsymbol{u}}^k$ closer to describing the movement of the RoI rather than the local movement of the block. Therefore, if the magnitude of $|\overline{\boldsymbol{u}}_i^j - \overline{\boldsymbol{u}}_i^k|, (f_i^j \in R_i^k)$ is larger than $T_{diff}$, which means a possible ME movement may occur, then $\tilde{\boldsymbol{u}}_i^j$ must be computed according to Eq. (8) no matter what state $f_i$ is classified as.

# 4 EXPERIMENTS

Table 1: Comparison with the state-of-the-art methods.

| Methods | CAS(ME)$^2$ | | | SAMM-LV | | | CASME II |
|---|---|---|---|---|---|---|---|
| | ME | MaE | All | ME | MaE | All | ME |
| MDMO(He et al., 2020) | 0.0082 | 0.1196 | 0.0376 | 0.0364 | 0.0629 | 0.0445 | - |
| SP-FD(Zhang et al., 2020) | 0.0547 | 0.2131 | 0.1403 | 0.1331 | 0.0725 | 0.0999 | - |
| OF-FD(Yuhong, 2021) | 0.1965 | 0.3782 | 0.3436 | 0.2162 | 0.4149 | 0.3638 | - |
| SOFTNet(Liong et al., 2021) | 0.1173 | 0.2410 | 0.2022 | 0.1520 | 0.2169 | 0.1881 | - |
| TSMSNet(Xue et al., 2021) | 0.1969 | 0.2395 | 0.2275 | 0.0878 | 0.2515 | 0.2466 | - |
| ABPN(Leng et al., 2022) | 0.1590 | 0.3357 | 0.3117 | 0.1689 | 0.3349 | 0.2908 | - |
| AUW-GCN(Yin et al., 2023) | 0.1538 | 0.4235 | 0.3834 | 0.1984 | **0.4293** | **0.3728** | - |
| S-pattern(Li et al., 2023b) | - | - | - | - | - | - | 0.4700 |
| **Ours** | **0.2614** | **0.5061** | **0.4558** | **0.2866** | 0.3724 | 0.3419 | **0.8571** |

We conduct the experiments on CASME II(Yan et al., 2014), CAS(ME)$^2$(Qu et al., 2018) and SAMM-LV(Yap et al., 2020) and follow the performance metrics proposed in MESNet (Wang et al., 2021). Appendix A.3 and A.4 introduce the details of datasets, metrics and experiment settings.

Table 1 reports our spotting results on three datasets together with the state-of-the-art(SOTA) methods. Our proposed MES algorithm outperforms the others on all three datasets. Each video in CASME II contains only one or zero ME and little head movement. We only perform step 1, 2, and 5 for MES in CASME II. The F1-score of 0.8571 and the recall of 0.8823 sufficiently demonstrate that the feature curve of ME conforms to the M-pattern under skip-$k$-frame calculation and M-pattern is more suitable for describing the ME movement in comparison to S-pattern (Li et al., 2023b). Except for MaE spotting results in SAMM-LV, the significant improvement in F1-score on CAS(ME)$^2$ and SAMM-LV is mainly attributed to the global movements elimination by block-wise MDMO in the feature extraction phase and the robustness and distinctiveness of M-pattern. The overall results indicate that the M-pattern applies to both macro and micro-expression spotting in long videos and our spotting rules are capable of distinguishing ME/ MaE from other movements. The main reason why our MaE spotting results in SAMM-LV is roughly 5% lower than the SOTA is that we have not extracted the RoIs below the eyes as proposed by He et al.Yuhong (2021), and these RoIs contain a considerable portion of MaEs related to blinking and eye movement.

We find that FP in ME spotting results are composed of three types of movements: habitual movements(with the highest proportion, such as eyebrow raising, pursing lips and pouting), MaE movements, and ME movements that do not satisfy Eq. (11). Habitual movements are almost identical with labeled ME in both magnitude and direction of movement. Thus, it is impossible to distinguish such habitual movements from ME using the unimodal data only. No global movements are detected in our results, which again demonstrates the superiority of our proposed methods in eliminating and distinguishing global movements. The detailed spotting results of our proposed methods are displayed in Appendix A.5.

**Parameter Analysis.** We focus on analyzing the effect of $T_{diff}$ on spotting results. $T_{diff}$ is the difference between the RoI MDMO $\overline{u}_i^j$ and block MDMO $\overline{u}_i^k$. It is designed to not compromise the possible ME movement when removing global movements based on the state of each frame. Fig. 9 illustrates the impact of varying $T_{diff}$ from 0 to 0.6 on the spotting results. $T_{diff} = 0$ means all the frames are treated as the order state. Hence no ME is compromised. In general, When $T_{diff} > 0$, we observe a large decrease in the number of FP for all spotting results in comparison to $T_{diff} = 0$ and the F1 scores have also improved. This is a solid evidence that block-wise MDMO can effectively eliminate global movements in the feature extraction phase. The number of TP (True Positives) at $T_{diff} = 0$ is almost equal to the number at $T_{diff} > 0$, which proves that by setting $T_{diff}$, we successfully preserve the possible micro-movements while eliminating the global movements. $T_{diff}$ is set to 0.4, 0.6, 0.2, 0.3, in the same order as the legend in Fig. 9. Setting a larger $T_{diff}$ might be able to eliminate more global movements, since most expressions have a magnitude exceeds 1.0. However, the larger $T_{diff}$ is, the more likely it is to disrupt the complete expression movements, especially for MaE, where the large magnitude of movement can easily cause the frame to be categorized as disorder state.

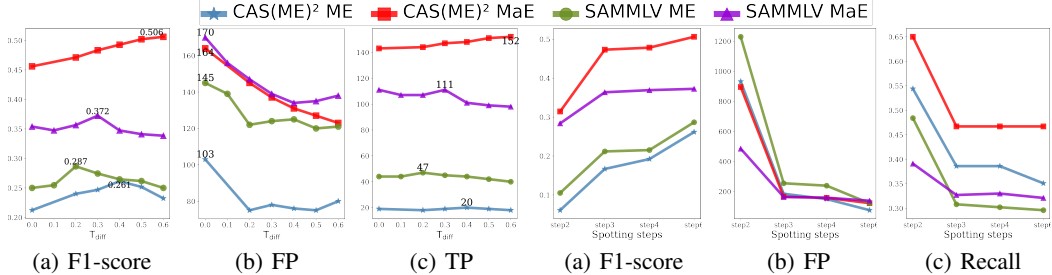

Figure 9: Spotting results for different $T_{diff}$.

Figure 10: Progressive spotting results across four spotting steps.

**Efficacy of spotting rules.** Fig. 10 illustrates the results from spotting step 2 to 6. Step 1 is designed to generate waves hence no spotting results are expected. We present the results after step 5(merging) for better analysis and visualization. As can be seen in Fig. 10 (a) and (b), each step is capable of eliminating FP according to a specific rule, leading to a gradual increase in F1-score. Step 3 and 6, spotting based on the characteristics of ME, contributes the most. It eliminates a considerable amount of FP that do not meet the ME characteristics, but at the same time, it also leaves TP with small magnitude out. The reason why step 6 is executed after merging is because we want to perform it from the perspective of a complete merged expression interval. The improvement by step 6 is more significant in ME than in MaE, because ME in CAS(ME)$^2$ and SAMM-LV have a distinct duration boundary, whereas MaE do not. The recall of SAMMLV MaE spotting at step 2 is the lowest as in Fig. 10 (c) and the overall performance is not as superior as the other three. The first reason has been explained above. The second reason is probably because larger $k(130)$ are less suitable for the proposed spotting algorithm than smaller $k$ (7, 18, 37, 34).

Table 2: Ablation experiments.

|  | CAS(ME)2 | | SAMM-LV | |
| --- | --- | --- | --- | --- |
|  | ME | MaE | ME | MaE |
| bwMDMO | **0.261** | **0.506** | **0.287** | **0.372** |
| bwMDMO, RoI-level | 0.233 | 0.486 | 0.278 | 0.363 |
| bwMDMO, w/o step3, 6 | 0.159 | 0.455 | 0.204 | 0.358 |
| bwMDMO, w/o step4 | 0.224 | 0.498 | 0.279 | 0.367 |
| bwMDMO, w/o FA | 0.200 | 0.430 | 0.231 | 0.353 |
| bwMDMO, w/o [FA, step4] | 0.190 | 0.449 | 0.229 | 0.348 |
| Fixed RF, w/o FA (Yuhong, 2021) | 0.117 | 0.291 | 0.153 | 0.279 |

bwMDMO: Block-wise MDMO. RoI-level: spotting at RoI-level. FA: Pixelmatch-based facial alignment.

Since facial expressions are the result of coordinated movements across multiple RoIs, the proposed spotting rules are based on the expression level. We conduct experiments at RoI-level with the same spotting rules and the results are shown in Table 2. Expression-level is more superior than RoI-level in all spotting results. For ME, spotting at expression-level can eliminate intervals with small magnitude that are likely to be generated by MaE or other movements. However, these interval are inclined to be kept if we spot at the RoI-level, for their magnitude are closely similar to ME.

**Efficacy of facial alignment.** We conduct experiments on the raw video to verify the performance of the proposed facial alignment algorithm. The results reflect the effectiveness of our alignment method. We also verify the advantage of the skip-$k$-frame strategy over the fixed RF strategy in the absence of facial alignment. Due to short time span, the Skip-$k$-frame strategy has already reduced global movements to a great extent at the very beginning, whereas the fixed RF strategy heavily relies on facial alignment to eliminate global movements.

## 5 CONCLUSION

In this paper, we propose a skip-$k$-frame block-wise MDMO feature for MES. Under skip-$k$-frame strategy, we substantiate a novel unique M-pattern for describing complete ME/ MaE movements. The spotting rules are designed based on M-pattern and ME traits. Block-wise MDMO can simultaneously extract features within RoIs and eliminate global movements. Dynamic updating of the RF method enables the proposed pixelmatch-based facial alignment algorithm to reduce jitter between frames and enhance alignment performance. Experimental results demonstrate the efficacy of the proposed methods.

ACKNOWLEDGMENTS

This work was funded in part by the National Key Research and Development Program of China under number 2019YFA0706200, in part by the National Natural Science Foundation of China grant under number 62222603, 62076102, and 92267203, in part by the STI2030-Major Projects grant from the Ministry of Science and Technology of the People's Republic of China under number 2021ZD0200700, in part by the Key-Area Research and Development Program of Guangdong Province under number 2023B0303030001, and in part by the Program for Guangdong Introducing Innovative and Entrepreneurial Teams (2019ZT08X214).

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

## A APPENDIX

### A.1 PSEUDO-CODE FOR FACIAL ALIGNMENT

The proposed facial alignment algorithm consists of Alignment and Reference Frame Update. Python package Pixelmatch (Mapbox, 2022) provides a function $pixelmatch$ to calculate the pixel difference between two images and return mismatched pixel number $m$. We choose the nose region as the matching box, and it is cropped according to the center coordinate, which is why there are 4 inputs in function $pixelmatch$. The alignment is adjusting the center coordinate $c_i$ of current frame, until the difference between the current matching box and the reference match box is minimal. The process of adjustment is based on the ideology of greedy algorithm, selecting the direction in which $m$ decreases the most as the current adjustment direction until $m$ is minimized.

---
**Algorithm 1** Alignment

---
**Input:** current frame and center $f_i, c_i$,reference frame and center $f_R, c_R$
**Output:** center coordinate $c_i^a$ after alignment, mismatch $m$
   $directions = [[0,0],[1,0],[0,1],[-1,0],[0,-1]]$
   $f_R = f_0, c_i = c_R, update = 1, list_m = [0]*5$
   $m = \text{mismatch}(f_i, f_R, c_i, c_R)$
   **if** $m < M_{min}$ **then**
     **return** $c_i, m$
   **end if**
   **while** $update \neq 0$ **do**
     **for** $i = 0$ to $\text{len}(directions)$ **do**
       $list_m[i] = \text{pixelmatch}(f_i, f_R, c_i + directions[i], c_R)$
     **end for**
     $update = \text{argmin}(list_m)$
     $c_i = c_i + directions[update]$
   **end while**
   **return** $c_i, list_m[0]$

---

The head posture or facial appearance is prone to change over a prolonged period of time, which is why we update the RF dynamically. However, it is critical to guarantee that the updated RF is perfectly aligned to ensure subsequent frames do not deviate. The cases of immediate update include $m = 0$ and $m > M_{max}$, indicating perfect alignment and large posture change respectively. $T_{du}$ is set for delayed update, and it can be adjusted according to the FPS (Frames per second) of the dateset. To reduce jitter, $count$ increases only when the center coordinate of current cutting box differs from the center coordinate of previous cutting box.

---
**Algorithm 2** Reference Frame Update

---
**Input:** current frame and center $f_i, c_i$,reference frame and center $f_R, c_R$
**Output:** updated reference frame and center $f_R, c_R$
   $f_R = f_0, c_R = c_0, count = 0, m_s = inf, f_{tmp}, c_{tmp}$
   $c_i, m_i = \text{Alignment}(f_i, f_R, c_i, c_R)$
   **if** $m_i == 0$ or $m_i > M_{max}$ **then**
     $f_R = f_i, c_R = c_i, count = 0, m_s = inf$
   **else**
     **if** $count < T_{du}$ **then**
       **if** $m_i \leq m_s$ **then**
         $f_{tmp} = f_i, c_{tmp} = c_i, m_s = m_i$
       **end if**
       **if** $c_i \neq c_{i-1}$ **then**
         $count = count + 1$
       **end if**
     **else**
       $f_R = f_{tmp}, c_R = c_{tmp}, count = 0, m_s = inf$
     **end if**
   **end if**
   **return** $f_R, c_R$

---

## A.2 VALIDATION OF M-PATTERN

CASME II(Yan et al., 2014), CAS(ME)$^2$(Qu et al., 2018) and SAMM-LV(Yap et al., 2020) are applied to the validation of M-pattern. For CASME II, there is only one or zero ME with little head movements in each video. Thus it is a perfect dataset for validation of M-pattern. In contrast, the videos in CAS(ME)$^2$ and SAMM-LV consist of mutiple type of movements, so we use these two datasets to verify the robustness and distinctiveness of M-pattern. Fig. 11 illustrates the feature curves of MEs on the above three datasets, with the ground-truth between the red lines. It can be observed that the feature curves for MEs of various lengths all conform to M-pattern.

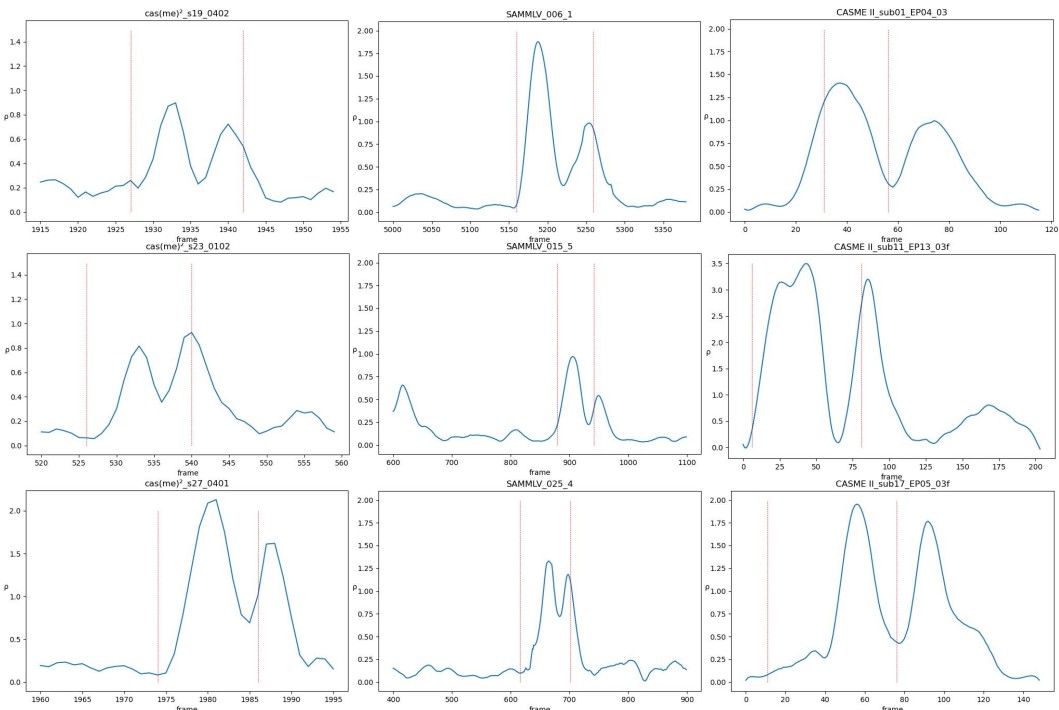

Figure 11: Feature curves of MEs in three datasets.

## A.3 DATASETS AND METRICS

**CAS(ME)$^2$** consists of 22 participants and 98 long videos, including 300 macro-expressions and 57 micro-expressions.

**SAMM-LV** consists of 32 participants and 147 long videos, including 343 macro-expressions and 159 micro-expressions in the long videos.

**CASME II** comprises of 246 micro-expressions out of 26 participants. Compared to the two datasets above, each video is shorter in duration and contains only one or zero micro-movement.

Table 3 summarizes the three databases.

We follow the performance metrics proposed in MESNet(Wang et al., 2021). For a spotted interval $W_{spotted}$, the prediction is considered as a true positive (TP) when there is a ground truth interval $W_{groundTruth}$ fitting the following condition:

$$\frac{W_{spotted} \bigcap W_{groundTruth}}{W_{spotted} \bigcup W_{groundTruth}} \geq 0.5. \tag{11}$$

Otherwise, it is false positive(FP). Specifically, if there are $m$ ground-truth intervals in dataset and $a$ ground-truth intervals are spotted; and if there are $n$ predicted intervals with $b$ TPs. Then the recall,

precision and F1-score are computed as follows:

$$Recall = \frac{a}{m}, Precision = \frac{b}{n}, F1 - score = \frac{2 \times Recall \times Precision}{Recall + Precision}. \tag{12}$$

The final evaluation is performed on the entire dataset, based on the overall F1-score of MaE and ME spotting performance.

Table 3: Details of CAS(ME)$^2$, SAMM-LV and CASME II

|  | CAS(ME)$^2$ | | SAMM-LV | | CASME II |
|  | ME | MaE | ME | MaE | ME |
|---|---|---|---|---|---|
| Frames per second | 30 | | 200 | | 200 |
| Num of samples | 57 | 300 | 159 | 343 | 255 |
| Max duration | 17 | 118 | 101 | 5175 | 141 |
| Min duration | 9 | 4 | 30 | 102 | 24 |
| $N$ | 14 | 36 | 74 | 259* | 67 |
| $k$ | 7 | 18 | 37 | 130* | 34 |

∗: For $N$, $k$ of SAMM-LV, samples over 800 frames are omitted.

## A.4 EXPERIMENT SETTINGS

We spot ME and MaE respectively using the proposed methods. TV-L1(Zach et al., 2007) is used for basic OF calculation. $p$ is set to 3 for peak detection. For MES, $T_{max} = \{23, 51\}, T_{ddeg} = \{45, 250\}, IoU_{merge} = \{0.3, 0.3\}(\{\cdot, \cdot\}$ for CAS(ME)$^2$ and SAMM-LV respectively). The MaE in datasets do not have discriminatory features like ME, thus we only set a minimum threshold $\{15, 9\}$ to exclude movements with lower $ddeg$ for MaE spotting. In feature extraction phase, $T_{disorder} = \frac{\text{block area}}{8 \times 8}$, where two 8s refer to 8 bins in total and the scaling factor respectively. Savitzky-Golay filter(Savitzky & Golay, 1964) is used to smooth the feature curve, where window length and polyorder is set to $k$ and 3. For facial alignment algorithm, frame with $m_i$ lower than $M_{min}$ will not execute Alignment in order to reduce jitter between frames; frame with $m_i$ larger than $M_{max}$ is likely to have a significant change in head posture or facial appearance. $M_{min}$ and $M_{max}$ is set to 0.1% and 8% of the matching box area respectively.

## A.5 EXPERIMENTAL RESULTS IN DETAIL.

**Detailed spotting results.** Table 4 shows the detailed spotting results. 'Find' refers to the number of spotted ME/MaE, while 'TP' refers to the number of predicted intervals that satisfy Eq. (11). Our approach drastically reduces the number of FP to about half of the total number of expressions (ME and MaE). Moreover, almost all FP were generated by habitual movements and no global movement is wrongly detected as ME/ MaE.

Table 4: Details of the spotting results.

|  | CAS(ME)$^2$ | | | SAMM-LV | | | CASME II |
|  | ME | MaE | All | ME | MaE | All | ME |
|---|---|---|---|---|---|---|---|
| Total | 57 | 300 | 357 | 159 | 343 | 502 | 255 |
| Find | 20 | 140 | 160 | 47 | 110 | 157 | 225 |
| TP | 20 | 152 | 172 | 47 | 111 | 158 | 225 |
| FP | 76 | 123 | 199 | 122 | 139 | 261 | 45 |
| Precision | 0.2083 | 0.5527 | 0.4636 | 0.2781 | 0.4440 | 0.3771 | 0.8333 |
| Recall | 0.3509 | 0.4667 | 0.4482 | 0.2956 | 0.3207 | 0.3127 | 0.8823 |
| F1-score | 0.2614 | 0.5061 | 0.4558 | 0.2866 | 0.3724 | 0.3419 | 0.8571 |

Table 5 provides the progressive results of spotting rules. Since the number of TP before merging far exceeds the number of ME, we adopt a new metric for better comparison. Compared to Eq. (12), the formula for $Precision$ is modified to $\frac{a}{n}$. Practically, the occurrence and restoration phases of ME

are not strictly symmetrical, hence two peaks in M-pattern are not equal in most cases. To prevent the smaller peak in M-pattern from being omitted, we set a lower threshold in step 1. Thus the initial number of FP is large. Leveraging the properties of ME (step 3, 6) helps us filter out a large number of FP. The merging (step 5) ensures that the merged interval is not shifted to the left or right and the length of merged interval is almost equal to the average length of intervals involved in merging. Only 2 MEs in SAMM-LV are excluded after merging for they no longer satisfy Eq.(11).

Table 5: Progressive MES results in detail.

| Step | CAS(ME)$^2$ ME | | | | SAMM-LV ME | | | |
|---|---|---|---|---|---|---|---|---|
| | Find | TP | FP | F1-score | Find | TP | FP | F1-score |
| 1, 2 | 32 | 74 | 2259 | 0.0600 | 88 | 319 | 3938 | 0.0399 |
| 1, 2, 3 | 22 | 35 | 293 | 0.1672 | 51 | 178 | 934 | 0.0803 |
| 1, 2, 3, 4 | 22 | 31 | 240 | 0.1765 | 50 | 175 | 838 | 0.0853 |
| 1, 2, 3, 4, 5 | 22 | 22 | 150 | 0.1921 | 48 | 48 | 239 | 0.2152 |
| 1, 2, 3, 4, 5, 6 | 20 | 20 | 76 | 0.2614 | 47 | 47 | 122 | 0.2866 |

In the absence of block-wise MDMO, the global movements can only be removed by spotting step 4. But the magnitude of some global movements are too small to be detected under skip-$k$-frame strategy. Therefore, as shown in Table 6, the experimental results using MDMO as a feature descriptor are worse than block-wise MDMO, mainly due to lower precision compared to the latter. We compare two different divisions of the OF field: intersecting division in Fig. 6(c) and disjoint division in Fig.7. For MDMO that only require magnitude feature, the difference between two divisions is not significant. Whereas for block-wise MDMO, which require directional information, the division proposed in this paper is noticeably superior to the disjoint division. Because both divisions average the OF vectors in the main direction, they are close in magnitude and differ more in direction. The proposed intersecting division method has a more accurate description in direction, as movement in any direction can ultimately be assigned to one bin.

Table 6: Ablation experiments.

| | CAS(ME)$^2$ | | SAMMLV | |
|---|---|---|---|---|
| | ME | MaE | ME | MaE |
| MDMO | 0.212 | 0.456 | 0.250 | 0.354 |
| MDMO* | 0.194 | 0.473 | 0.245 | 0.351 |
| MDMO, w/o step3, 6 | 0.155 | 0.431 | 0.197 | 0.345 |
| MDMO, w/o step4 | 0.180 | 0.455 | 0.249 | 0.352 |
| MDMO, w/o FA | 0.189 | 0.433 | 0.224 | 0.346 |
| MDMO, w/o [FA, step4] | 0.180 | 0.439 | 0.222 | 0.343 |
| bwMDMO | 0.261 | 0.506 | 0.287 | 0.372 |
| bwMDMO* | 0.200 | 0.486 | 0.241 | 0.345 |

MDMO: All frames are treated as order state, i.e., $T_{diff}$ is set to 0. bwMDMO: Block-wise MDMO. FA: Pixelmatch-based facial alignment. *: Original partition of 8 bins(Fig. 7).

