# OpenReview forum: "A unique M-pattern for micro-expression spotting in long videos"
_ICLR.cc/2024/Conference — ICLR 2024 poster_

### Official Review · Reviewer_32Bs · 2023-10-31

**Soundness:** 3 good
**Presentation:** 3 good
**Contribution:** 2 fair
**Rating:** 6
**Confidence:** 2

**Summary:**

This article focuses on the issue of global movements affecting the recognition of subtle micro-expressions in micro-expression analysis. It identifies a distinctive motion pattern in micro-expressions, referred to as the "M pattern," and introduces the feature extraction method "skip-k-frame block-wise main directional mean optical flow (MDMO)" based on this M pattern. And propose a dynamic update facial alignment algorithm based on pixel matching with unfixed reference frames. The proposed methods are superior to the state-of-the-art methods.

**Strengths:**

1. This article identifies a unique motion pattern (M-pattern) in micro-expressions and proposes a feature extraction method based on the M-pattern that promises to address the effects of global motion.
2. Compared to other methods discussed in the article, this approach has achieved the current state-of-the-art results in Micro-Expression Spotting tasks and provides partial explanations for the issues it aims to address.

**Weaknesses:**

1. No more detailed theoretical analyses of why the M-pattern is effective in mitigating the influence of other global movements. It also does not offer the necessary alternative modes or direct ablation experiments involving muscle intensity.

**Questions:**

The method in the article appears to establish a set of conditions and rules, and I am skeptical about whether this is a universally applicable approach in this field. Is it possible to demonstrate through experimentation that this method exhibits generalizability on other datasets?

---

> ### Author Response · Authors · 2023-11-20
> **Response to Reviewer 32Bs (1/2)**
>
> Thank you for the valuable comments. In the following, your comments are first stated and then followed by our point-by-point responses.
>
> ---
>
> **[Q1]** No more detailed theoretical analyses of why the M-pattern is effective in mitigating the influence of other global movements.
>
> **[A1]** In long videos, changes in head posture, image background, brightness, and other factors become more and more apparent over time. Therefore, an obvious drawback of mainstream spotting algorithms based on fixed reference frame is that frames distant from the reference frame are subject to interference from these changes, resulting in the inability to accurately extract subtle micro-expression features.
>
> The M-pattern is effective in mitigating the influence of global movements for two main reasons: (1) The M-pattern is derived from the proposed skip-$k$-frame (unfixed reference frame strategy) feature extraction, i.e., computing the feature between frame $i$ and frame $i + k$. The two frames involved in optical flow computation are only $k$ frames apart, minimizing changes in  head posture, brightness and etc. with much shorter time span. This allows us to focus on changes predominantly originating from facial muscle movements within RoIs. (2) The M-pattern is derived from the unique movement pattern of micro-expressions. It contains information about the occurrence phase, restoration phase, intensity and duration of micro-expressions. This information can clearly distinguish between micro-expressions and global movements, since the properties of global movements (global, prolong, large-magnitude) are almost opposite to those of micro-expressions (localized, transient, small-magnitude).
>
> We apologize for neglecting to analyze in depth why the M-pattern is effective in mitigating the influence of other global movements and will refine the relevant content in our future versions.
>
> ---
>
> **[Q2]** It also does not offer the necessary alternative modes or direct ablation experiments involving muscle intensity.
>
> **[A2]** Thank you for bringing this issue to our attention. The existing methods quantify muscle intensity simply by using the peaks of the feature curve. However, for videos with a low frame rate and frequent global movements, the overall feature curve will exhibit significant fluctuations, and the feature value will be relatively larger. Therefore, we quantify “muscle intensity” in terms of multiples, defined as muscle deformation degree $ddeg$:
> $$
> ddeg_i^j=c_i^j/avg_i^j,
> $$
> where $avg_i^j$ is the average of $3*fps$ frames preceding the start frame of predicted expression $e_i^j$. $c_i^j$ is the peak value of  $e_i^j$. Two indicators are used to distinguish between micro-expressions and interference movements (mainly macro-expression movements and global movements): threshold $T$ (Section 3.3, step 1) and muscle deformation degree $ddeg$. The former represents a coarse selection, while the latter represents a fine selection. We add a direct ablation experiment on $ddeg$. As shown below, we verify the effectiveness of $ddeg$ without merging (step 5), which gives a more intuitive analysis.
> | FIND/FP/F1 |    $~~~~~$CAS(ME)$^2$   |    $~~~~$SAMM-LV    |
> |:----------:|:-------------:|:-------------:|
> |   w/ $ddeg$  | 25/128/0.2264 | 47/406/0.1322 |
> |  w/o $ddeg$  | 25/169/0.1909 | 47/450/0.1245 |
>
> The introduction of $ddeg$ reduces a portion of FPs while retaining all TP intervals. $ddeg$ can differentiate between macro and micro-expressions at a finer granularity.
>
> In addition, we describe how to better utilize “muscle internsity” to distinguish micro-expressions in Section 3.3, step 3. We utilize $ddeg$ to spot ME at the expression-level, rathen than at the RoI-level because the expression are the result of coordinated movements across multiple RoIs. The experimental results in Section 4 confirm that spotting at the expression-level is a better choice.
>
> ---
>
> **[Q3]** The method in the article appears to establish a set of conditions and rules, and I am skeptical about whether this is a universally applicable approach in this field.
>
> **[A3]** YES. It is a universally applicable approach in the field of MES to establish a set of conditions and rules, primarily aimed at reducing the number of false positives. Different methods will design different rules according to their own characteristics. Our method is designed strictly based on the characteristics of micro-expressions, which has achieved favorable results.

---

> ### Author Response · Authors · 2023-11-20
> **Response to Reviewer 32Bs (2/2)**
>
> **[Q4]** Is it possible to demonstrate through experimentation that this method exhibits generalizability on other datasets?
>
> **[A4]** YES, it is indeed possible to demonstrate through experimentation that our method exhibits generalizability on other datasets. MES in long videos has only gained attention in recent years, and there are relatively few available datasets. Following the guidelines in Facial Micro-expression Grand Challenge (MEGC), almost all researchers use CAS(ME)$^2$ or SAMM-LV or both to validate their methods. Two new spontaneous micro-expression datasets MMEW and CAS(ME)$^3$ were proposed in 2021 and 2022, respectively. But currently, few works utilize these two datasets; and they are more commonly employed in micro-expression recognition task. We will conduct experiments on these two datasets to validate the generalizability of our method in future works.
>
> In addition, in our experiments, we added CASME II to validate our method and yielded 0.8571 F1-score. Although CASME II contains fewer global movements, the experimental results are sufficient to illustrate the superiority of our method in describing, quantifying, and spotting micro-expressions.

---

> ### Comment · Reviewer_32Bs · 2023-11-22
> **Review of author responses**
>
> Dear authors, area chair and fellow reviewers,
>
> After careful consideration of the rebuttal provided by the authors and other reviewers of this paper, I have decided to accept this paper.
> I believe the authors have addressed the initial concerns and that their paper will be a meaningful addition to the conference proceedings. And I'm going to keep my original judgment.

---

### Official Review · Reviewer_KsQW · 2023-11-01

**Soundness:** 1 poor
**Presentation:** 1 poor
**Contribution:** 2 fair
**Rating:** 5
**Confidence:** 2

**Summary:**

This paper reported a “M” shaped pattern of optical flow magnitude curve that could be used in micro-expression spotting. Based on the M-pattern assumption, the author proposed a three stage, rule based system to estimate micro-expression interval. The author further compared the proposed method with other micro-expression spotting methods and obtained favorable performance on public dataset.

**Strengths:**

1. The proposed method has superior performance on important public datasets compared to several other micro-expression spotting methods.

**Weaknesses:**

1. While the proposed system has many parameters, it is not clear how to choose the parameter value.
2. There isn’t necessary discussion on the type of micro-expression presented in the dataset, as well as how well the proposed M-pattern fit in each category.
3. In general I think the structure of the paper could be improved. For example, putting dataset description in appendix makes the experiment section harder to follow.

**Questions:**

1. For different type of micro-expression spotting, does the system need different set of parameters?
2. Does the dataset contain a validation part for parameter selection, and a separate test part for metric reporting?

---

> ### Author Response · Authors · 2023-11-20
> **Response to Reviewer KsQW (1/2)**
>
> We thank Reviewer KsQW for the valuable comments and suggestions. Below we address your comments and questions.
>
> ---
>
> **[Q1]** While the proposed system has many parameters, it is not clear how to choose the parameter value.
>
> **[A1]** We apologize for not clearly explaining how to select the parameter value. It is a universally applicable approach in the field of micro-expression spotting (MES) to establish a set of rules and parameters to reduce the number of false positive intervals, mainly resulting from global movements. Our method establishes rules and parameters completely based on the characteristics of micro-expression. Below we explain in detail how the parameters are selected.
>
> - **$M_{min}$** and **$M_{max}$**.
>
> **[Role]** In facial alignment stage,  we use the number of mismatched pixels between the current frame and the reference frame to align the face. Consider two extreme cases: (1) if the alignment is performed on every frame, then successive frames will be very jittery; (2) if all frames are aligned to the same reference frame, then it will not be possible to align them when a significant change in head pose or facial appearance occurs. Therefore, $M_{min}$ is set to reduce the jitter between images and $M_{max}$ is a threshold that implies a significant change in head pose or facial appearance.
>
> **[How to choose]** We chose them randomly within a certain range, because through multiple experiments, we found that the alignment performance and the spotting results are not sensitive to these two parameters due to the superiority of our proposed method in eliminating global movements. $M_{min}$ and $M_{max}$ are set to 0.1% and 8% of the cutting box area, respectively. Specifically, (1) $M_{min}$ is chosen from [0.05%, 0.1%]. Smaller $M_{min}$ results in continuous jitter between frames, and larger values results in abrupt changes between frames. (2) $M_{max}$ is chosen from [5%, 10%]. While the definition of “significant change” is ambiguous, we can determine that a “significant change” must correspond to a relatively large mismatch value in terms of pixelmatch.
>
> - **$IoU_{merge}$**.
>
> **[Role]** Expression is the result of coordinated movements across multiple RoIs (Regions of Interest). For example, a smiling expression may involve movements of the mouth and eyebrow regions. We performed MES on 12 facial RoIs. Each RoI generates several predicted micro-expression intervals. Intervals with $IoU$ (Intersection over Union) larger than $IoU_{merge}$ will be merged since they belong to the same expression.
>
> **[How to choose]** We utilize a grid search with range [0.2, 0.9] and a stride of 0.05 to determine its value.
>
> - **$T_{ddeg}$** and **$T_{max}$**.
>
> **[Role]** $T_{ddeg}$ and $T_{max}$ are proposed to distinguish between micro-expressions and interference movements. Interference movements are the primary cause of low spotting accuracy, mainly including global movements (head rotation, image blurring...) and macro-expression movements. From the perspective of RoI, the magnitude of global movements and macro-expressions on certain RoIs is close to the magnitude of micro-expressions, making it challenging for existing methods to differentiate between micro-expressions and global movements. However, from the perspective of expression, The overall magnitude of interference movements is significantly larger than the overall magnitude of of micro-expressions. Therefore, we propose $T_{ddeg}$ and $T_{max}$ to spot micro-expression at the expression-level.
>
> **[How to choose]** We first conducted a coarse grid search with stride 10, followed by a finer search with a smaller stride.
>
> ---
>
> **[Q2]** In general I think the structure of the paper could be improved. For example, putting dataset description in appendix makes the experiment section harder to follow.
>
> **[A2]** Thank you for bringing this issue to our attention. We all agree with your constructive comment that the structure of this paper needs improvement and apologize for any confusion caused by the inappropriate structure. We placed the dataset description in appendix due to limitations in the length of the paper. We will surely refine the structure of the paper in our future versions.

---

> ### Author Response · Authors · 2023-11-20
> **Response to Reviewer KsQW (2/2)**
>
> **[Q3&4](Questions regarding the type of micro-expression)** There isn’t necessary discussion on the type of micro-expression presented in the dataset, as well as how well the proposed M-pattern fit in each category. For different type of micro-expression spotting, does the system need different set of parameters?
>
> **[A3&4]** We apologize for not explaining the micro-expression spotting (MES) task clearly in paper. As a matter of fact, we had not discuss the type of micro-expression in paper. Similar to temporal action localization, we discuss locating all the micro-expression intervals in a long video, i.e., determine the onset frame (the start of expression) and offset frame (end of expression) of the micro-expression. We don't focus on **WHAT** category micro-expression belongs to; we focus on **WHEN** micro-expression occurs. According to the guidelines presented in Facial Micro-expression Grand Challenge (MEGC) [1], we spot two types of expressions in long videos, macro-expression and micro-expression. The expressions we refer to in our daily life are macro-expression, which include prolonged, large-magnitude, global facial muscle movements such as laughing and crying. The expression that unconsciously leaks out when we attempt to conceal our emotions is referred to as micro-expression. Opposite to macro-expression, micro-expression is characterized by transient, small-magnitude, localized. For this reason, micro-expression and macro-expression are spotted separately.
> The proposed M-pattern is used to characterize the movement of micro-expressions, includes information about the occurrence phase, restoration phase, intensity and duration of micro-expressions. And it is also applicable to describing macro-expressions. The experimental results demonstrate the superiority of our method for both micro-expression spotting and macro-expression spotting.
>
> For micro-expression spotting and macro-expression spotting, the system needs different set of parameters, which is mentioned in Appendix A.4. Micro-expression and macro-expression have completely opposite properties and no consensus has yet been reached on the definition of these two types of expressions. It is difficult to distinguish between macro and micro-expression without designing effective parameters.
>
> ---
>
> **[Q5]** Does the dataset contain a validation part for parameter selection, and a separate test part for metric reporting?
>
> **[A5]** As mentioned above, MES aims to locate intervals in long videos where micro-expressions occur. Unlike recognition task that focus on the category of micro-expressions, spotting task concentrates on when micro-expressions occur and predicts the onset (the start of expression) and offset (the end of expression) frames of micro-expression. However, the presence of a large amount of global movements in long videos severely interferes with the spotting of micro-expressions. Therefore, our work focuses on eliminating global movements as well as proposing a unique micro-expression movement pattern for better spotting. We follow the performance metric proposed in Facial Micro-expression Grand Challenge (MEGC) [1]. For a spotted interval $W_{spotted}$, the prediction is considered as a true positive (TP) when there is a ground truth interval $W_{groundTruth}$ fitting the following condition:
>
> $$
> \frac{W_{spotted} \bigcap W_{groundTruth}}{W_{spotted} \bigcup W_{groundTruth}} \geq 0.5.
> $$
>
> Otherwise, it is false positive(FP). Specifically, if there are $m$ ground-truth intervals in dataset and $a$ ground-truth intervals are spotted; and if there are $n$ predicted intervals with $b$ TPs, then the recall, precision and F1-score are computed as follows:
>
> $$
> Recall=\frac{a}{m}, Precision=\frac{b}{n},
> F1-score=\frac{2\times Recall \times Precision}{Recall+Precision}.
> $$
>
> Traditional methods test spotting performance directly on the entire dataset and do not divide the dataset into train, validation and test sets. Currently, tradition methods are able to effectively distinguish micro-expressions from global movements by utilizing prior knowledge in the field of micro-expressions, and the spotting results are comparable to, or even better than, most of the deep-learning methods, which have relatively poor spotting results due to the excessive number of false positive intervals.
>
> ---
>
> **Reference**
>
> [1] Li, J. , Wang, S. J. , Yap, M. H. , See, J. , & Li, X. . (2020). MEGC2020 -The Third Facial Micro-Expression Grand Challenge. 2020 15th IEEE International Conference on Automatic Face and Gesture Recognition (FG 2020) (FG). IEEE.

---

> ### Comment · Reviewer_KsQW · 2023-11-23
> **Response to the authors's comment**
>
> Thanks the authors for the clear responses and clarifications. Given the clarity and thoughtfulness of the response, I am please to update the rating to: 5: marginally below the acceptance threshold.

---

### Official Review · Reviewer_Z3Ax · 2023-11-07

**Soundness:** 3 good
**Presentation:** 3 good
**Contribution:** 3 good
**Rating:** 8
**Confidence:** 4

**Summary:**

In this paper, the authors address the challenging task of micro-expression(ME) spotting, which is difficult due to its subtle nature and other factors, such as head rotation, a global movement during an ME generation. A novel approach called skip-k-frame block-wise main directional mean optical flow (MDMO) feature is proposed by the authors for ME spotting. This is achieved using unfixed reference frames compared to existing work utilizing fixed reference frames. The paper introduces the readers to M-pattern, a distinct movement pattern and enhanced spotting rules based on M-pattern and ME characteristics. The article then presents block-wise MDMO features to remove global movements without compromising ME movements in the feature extraction stage. Experimental results on state-of-the-art datasets demonstrate that the proposed methods outperform existing methods in ME spotting.

**Strengths:**

The paper highlights the strength of recognizing patterns exhibited by actions and further utilizing characteristics exhibited by those actions in spotting them effectively. The authors have validated this understanding in the domain of ME spotting. The proposed work is an improvement on the existing work in the literature.

**Weaknesses:**

Even though it is a well-written paper, the draft contains many typos. It is of utmost importance that these are fixed.

Some aspects were unclear to the reviewer and are listed under questions.

**Questions:**

The reviewer would like to get feedback on the following:
1. The k value in skip-k-frames was calculated as (N+1)/2, where N is the average ME duration. How does N (average value) influence the spotting of MEs that fall away (beyond a threshold) from this mean? Ex. In Fig 11, was the spotting of specific MEs easier than others (CASME II_sub11 type compared to CASME II_sub01 type)?

2. It was unclear to the reviewer if the authors updated values of k across datasets (based on the average ME duration calculated from the chosen sample set).

3. If indeed k was updated as per the sample set, how would this impact its update in the case of very long videos (which were omitted by authors of this work)?

---

> ### Author Response · Authors · 2023-11-20
> **Response to Reviewer Z3Ax**
>
> We thank Reviewer Z3Ax for the good words and constructive suggestions!
>
> ---
>
> **[Q1]** Even though it is a well-written paper, the draft contains many typos. It is of utmost importance that these are fixed.
>
> **[A1]** We apologize for typos! We have double-checked for any spelling and grammar errors carefully, and improved the presentation of some content. Part of the corrected sentence is displayed as follows:
>
> (1)Based on the limitations above, we adopt **an** unfixed RF strategy for both facial alignment and feature extraction.
>
> (2)Yan et al. (2014) **utilize** Local Weighted Mean(LWM) transformation to...
>
> (3)Common feature descriptors include MDMO (Liu et al., 2016), LCN (Yap et al., 2022), LBP-TOP(Tran et al., 2019), HOOF (Verburg & Menkovski, 2019)**, etc.**
>
> (4)Third, based on M-pattern and ME characteristics, we **provide novel ME spotting rules**.
>
> (5)But we must ensure that the updated RF is well-aligned to prevent subsequent frames **from aligning with a misaligned frame**.
>
> (6)... satisfy the assumptions of OF (minor displacement and constant apparent brightness) even without facial **alignment**.
>
> (7)Thus, we **propose spotting rules** based on M-pattern and ME characteristics.
>
> ---
>
> **[Q2]** The $k$ value in skip-$k$-frames was calculated as ($N$+1)/2, where $N$ is the average ME duration. How does $N$ (average value) influence the spotting of MEs that fall away (beyond a threshold) from this mean? Ex. In Fig 11, was the spotting of specific MEs easier than others (CASME II_sub11 type compared to CASME II_sub01 type)?
>
> **[A2]** From the perspective of duration, in Fig. 11, the duration of CASME II_sub11_EP13_03f (76 frames) is longer than $N$ (67 frames) and the duration of CASME II_sub01_EP04_03 (26 frames) is much shorter than $N$. However, the proposed M-pattern is applicable to micro-expressions of various lengths. In section 3.2, we described how a unique M-pattern is obtained for a micro-expression of length $N$. For MEs that fall away (beyond a threshold) from $N$, we elaborate on the corresponding spotting rules in step 2 of Section 3.3. Therefore, the spotting results does not deviate due to different durations. From the perspective of the shape of M-pattern, although the M-pattern in CASME II_sub11_EP13_03f looks relatively distorted compared to CASME II_sub01_EP04_03, there is no difference in spotting these two MEs as we focus on the overall trend. In summary, the influence of $N$ on MEs that fall away (beyond a threshold) from mean value is not significant.
>
> ---
>
> **[Q3&4]** It was unclear to the reviewer if the authors updated values of $k$ across datasets (based on the average ME duration calculated from the chosen sample set). If indeed $k$ was updated as per the sample set, how would this impact its update in the case of very long videos (which were omitted by authors of this work)?
>
> **[A3&4]** We apologize for any confusion caused by our unclear presentation. The value of $k$ is updated across datasets (Appendix A.3). The $k$ value in skip-$k$-frame is calculated as ($N$+1)/2, where $N$ is the average duration of all micro-expressions in the dateset. For datasets CAS(ME)$^2$, SAMM-LV, CASME II, k is set to 7, 37, 34, respectively. The details are shown as follows.
>
> |   |   $~~$CAS(ME)$^2$|  SAMM-LV  | CASME II |
> |:-:|:---------:|:---------:|:--------:|
> |   | $~~$ME \| MaE | ME \| MaE |    ME    |
> | $N$|14 \| 36 | 74 \| 259 |    67    |
> | $k$ | $~~$7 \| 18 | 37 \| 130 |    34    |
>
> For each dataset, k is fixed, and all videos in the dataset use the same k value, regardless of the duration. Therefore, $k$ is not updated per sample set, so it does not affect the spotting results in the case of very long videos.

---

> > ### Comment · Reviewer_Z3Ax · 2023-11-22
> > **Review of author responses**
> >
> > Dear authors,
> >
> > Thank you for taking the time and drafting clear responses.
> >
> > Dear area chairs and Reviewers,
> >
> > Given the feedback from the authors, I am pleased to stand by my initial rating.
> >
> > Thanks

---

### Official Review · Reviewer_yjCN · 2023-11-07

**Soundness:** 3 good
**Presentation:** 3 good
**Contribution:** 3 good
**Rating:** 8
**Confidence:** 3

**Summary:**

The paper presents a skip-k-frame strategy for micro expressions spotting in videos. They present a Main Directional Mean Optical Flow (MDMO) feature that is capable of removing global movements from feature extraction stage. Experimental results were presented on three popular datasets and comparison against state of the art methods.

**Strengths:**

- Elimination of the global movements in the early feature extraction stage contributes to better detection of micro expressions.
- The paper is well written and the proposed methodology seem to be technically sound
- The experimental results show that the proposed method mostly outperformed the state of the art methods

**Weaknesses:**

I did not find much weaknesses in the paper

**Questions:**

On a minor note I have following suggestions to improve the paper:
- In section 3.1 Face Alignment, how do you determine the values for M_min and M_max?
- Is it possible to show one visual example of elimination of global movements from feature extraction stage?
- Some of the graphs are very hard to read in a printed paper as the font size is so small, for example Figures 9 and 10

---

> ### Author Response · Authors · 2023-11-19
> **Response to Reviewer yjCN**
>
> We thank Reviewer yjCN for the good words and constructive suggestions!
>
> ---
>
> **[Q1]** In section 3.1 Face Alignment, how do you determine the values for $M_{min}$ and $M_{max}$?
>
> **[A1]** $M_{min}$ is set to reduce the jitter between images and $M_{max}$ is a threshold that implies a significant change in head pose or facial appearance. $M_{min}$ and $M_{max}$ are set to 0.1% and 8% of the cutting box area respectively, which we have mentioned in Appendix A.4. But we did not discuss why we chose 0.1% and 8%. We apologize for any confusion caused to our readers and will refine the relevant content in our future versions. As a matter of fact, we randomly chose these two parameters within a certain range because through multiple experiments, we found that the alignment performance and the spotting results are not sensitive to these two parameters. Specifically, (1) $M_{min}$ is chosen from [0.05%, 0.1%]. Smaller $M_{min}$ results in continuous jitter between frames, and larger $M_{min}$ values results in abrupt changes between frames. (2) $M_{max}$ is chosen from [5%, 10%]. While the definition of “significant change” is ambiguous, we can determine that a “significant change” must correspond to a relatively large mismatch value in terms of pixelmatch. Due to the proposed algorithm's robustness and its superiority in eliminating global movements, changes in these two parameters will not lead to significant variations in spotting results.
>
> ---
>
> **[Q2]** Is it possible to show one visual example of elimination of global movements from feature extraction stage?
>
> **[A2]** YES. We can demonstrate how the proposed methods eliminate global movements in feature extraction stage by visualizing and comparing the feature curves computed by block-wise MDMO and MDMO respectively. We will add the visualized curves in our future versions.
>
> ---
>
> **[Q3]** Some of the graphs are very hard to read in a printed paper as the font size is so small, for example Figures 9 and 10.
>
> **[A3]** Thanks for bringing this issue to our attention. We will recreate these images to ensure clarity for reading on printed paper.

---

### Official Review · Reviewer_o3nk · 2023-11-08

**Soundness:** 3 good
**Presentation:** 2 fair
**Contribution:** 3 good
**Rating:** 8
**Confidence:** 2

**Summary:**

The paper introduces a novel mean-optical-flow (MDMO) feature for Micro-expression spotting, along with a series of spotting rules on the M-curve yielded by such features.
The authors were successful in showing how Pixelmatch-based facial alignment followed by MDMO feature extraction are effective to hamper disturbances from global movements that greatly impact the performance of MES methods. They also provided enough evidence of the effectiveness and soundness of the proposed spotting rules.
The authors claim to have outperformed SotA methods and provide numerical results to sustain their claim.

**Strengths:**

The authors were successful in clearly describing the importance of the MES task, the state of the art and, more importantly, the limitations of the field, from a technical point of view.
The proposed method, in my opinion, is adequate to sidestep such limitations and the experimental setup. Both MDMO and the M-curve analysis are interesting choices, the the experimental setup, including the comparison to related methods and ablation studies, is clearly well designed to not leave gaps when justifying both the effectiveness of the method, and the design choices.

**Weaknesses:**

They are exclusively related to the presentation of the paper.
In first place, I think the order of Section 3.4 should be before Section 3.2, to match the order of the steps inside the pipeline.
On the other hand, the spotting rules are presented in a really verbose manner, making Section 3.3 denser than it should be. I think that the authors can rather present the details in a graphical way (probably a flowchart) and dedicate the main text to describe the decision pipeline in more plain words.
Following the guidelines of other top-tier conferences, I think it is necessary the authors dedicate a few lines on ethical concerns implied in automating MES spotting. This field is particularly sensible from a bias and fairness perspective.
Finally, I spotted some typos:

Page 3
- Alignment is the process of adjusting the center coordinate until m between the current matching box and the reference matching box is minimized.
- to prevent subsequent frames from being aligned with misaligned frame
- even without facial alignmen.
- Thus, we propose a spotting rules

**Questions:**

- Is there any particular reason to present the feature extraction procedure before describing the M-pattern? (Of course, I can have misunderstood the order of the steps in the method and feature extraction foes last indeed.)
- You get a unique M-curve for the whole sequence of is one M-curve per ROI?

**Details Of Ethics Concerns:**

The particular field of Spontaneous micro-expression, to the best of my knowledge, is a very controversial one nowadays, as there is no consensus about its effectiveness as indicator of emotions that individuals attempt to restrain.
Although the problem, from a computer vision perspective is, with not doubts, interesting, using technology to automate the task of MES could encourage inappropriate the use of ME in criminal investigation or psychology.

---

> ### Author Response · Authors · 2023-11-19
> **Response to Reviewer o3nk (1/2)**
>
> Thank you for the valuable comments. We address your comments and questions below.
>
> ---
>
> **[Q1]** Is there any particular reason to present the feature extraction procedure before describing the M-pattern? (Weakness: I think the order of Section 3.4 should be before Section 3.2, to match the order of the steps inside the pipeline.)
>
> **[A1]** The MES process proceeds as follows: (1) facial alignment, (2) feature extraction, (3) spotting. Following the pipeline, the order of Section 3 should be 3.1, 3.4, 3.2, 3.3. The spotting rules are performed on the feature curves obtained in the feature extraction phase. However, we placed feature extraction in Section 3.4 and introduced M-pattern at first. We adopted this writing order for several reasons: (1) the M-pattern was derived from the computation of skip-$k$-frame, and the strategy of unfixed reference frame (skip-$k$-frame) was used throughout the paper; (2) we inferred M-pattern first (Section 3.2), followed closely by the introduction of M-pattern-based spotting rules (Section 3.3). However, the spotting rules cannot address global movements that exhibit single-wave with similar magnitudes to ME on the feature curve. Therefore, we must conduct a global analysis in the early feature extraction stage to eliminate such global movements (Section 3.4).
>
> We have considered the possibility that such an arrangement might cause distress to the reader when writing the paper. And we will reorganize the order and revise the relevant content to make it consistent with the pipeline in future revisions.
>
> ---
>
> **[Q2]** The spotting rules are presented in a really verbose manner, making Section 3.3 denser than it should be. I think that the authors can rather present the details in a graphical way (probably a flowchart) and dedicate the main text to describe the decision pipeline in more plain words.
>
> **[A2]** We thank the reviewer for the constructive suggestions. The spotting rules proposed in this paper are quite different and novel compared to existing methods. In step 2, we provided a detailed explanation on how to utilize the M-pattern to detect ME of varying durations. We also elaborated on spotting of ME without a restoration phase, thereby enhancing the overall comprehensiveness and robustness of the spotting capability. In step 3, we introduced a novel measurement muscle deformation degree $ddeg$ to describe muscle intensity and explain in detail how to reduce FPs at the expression-level, rather than at the RoI-level as in existing methods. As a result, Section 3.3 has an extended length. Thank you for the constructive suggestions and we will illustrate the spotting rules in the most intuitive way using graph and refine the content of Section 3.3.
>
> ---
>
> **[Q3]**(Details Of Ethics Concerns): Following the guidelines of other top-tier conferences, I think it is necessary the authors dedicate a few lines on ethical concerns implied in automating MES spotting. This field is particularly sensible from a bias and fairness perspective.
>
> **[A3]** Thank you for bringing this issue to our attention. Indeed, there are potential ethics concerns in the field of micro-expression. The micro-expression community is doing its best to protect the privacy of the subjects and to avoid inappropriate use of micro-expression information. First, to the best of our knowledge, all spontaneous micro-expression datasets adhered to ethical guidelines and were approved by the review board. The subjects are clearly informed of all relevant matters and their rights. Second, micro-expression analysis is significant in various fields, including psychology, medicine, criminal investigation, and security. At present, micro-expression analysis is mainly performed manually by specialists, and automated micro-expression spotting technology is not mature enough to be used in real-world scenarios. Last but not least, regardless of the means, we strongly advocate for the correct and lawful use of this technology.
>
> The following is the content regarding ethics concerns that will be included in the article: “Currently, research on automated MES is still in a lab-controlled scenario. However, potential ethics concerns remain, including leakage of sensitive personal information, illegal or excessive access to others' expression information, and inappropriate use in specific fields such as criminal investigation and psychology. In future, substantial efforts need to be invested to address these concerns.”

---

> ### Author Response · Authors · 2023-11-19
> **Response to Reviewer o3nk (2/2)**
>
> **[Q4]** I spotted some typos.
>
> **[A4]** We apologize for typos! We have corrected the typos carefully and refined relevant contents. Part of the corrected sentence is displayed as follows:
>
> (1)Based on the limitations above, we adopt **an** unfixed RF strategy for both facial alignment and feature extraction.
>
> (2)Yan et al. (2014) **utilize** Local Weighted Mean(LWM) transformation to...
>
> (3)Common feature descriptors include MDMO (Liu et al., 2016), LCN (Yap et al., 2022), LBP-TOP(Tran et al., 2019), HOOF (Verburg & Menkovski, 2019)**, etc.**
>
> (4)Third, based on M-pattern and ME characteristics, we **provide novel ME spotting rules**.
>
> (5)But we must ensure that the updated RF is well-aligned to prevent subsequent frames **from aligning with a misaligned frame**.
>
> (6)... satisfy the assumptions of OF (minor displacement and constant apparent brightness) even without facial **alignment**.
>
> (7)Thus, we **propose spotting rules** based on M-pattern and ME characteristics.
>
> ---
>
> **[Q5]** You get a unique M-curve for the whole sequence or is one M-curve per ROI?
>
> **[A5]** In the task of MES in long videos, each video contains several or zero micro-expressions, macro-expressions, along with a considerable amount of global and habitual movements. Therefore, for a video, we get several or zero M-curves (predicted intervals) per RoI, and intervals with IoU (Intersection over Union) larger than $T_{merge}$ will be merged since they belong to the same expression.

---

> > ### Comment · Reviewer_o3nk · 2023-11-20
> > **Rating Update**
> >
> > Dear authors, area chair and fellow reviewers,
> >
> > After checking the authors' answer and the interesting discussion they had with the other reviewers, I'm pleased to inform that I updated my recommendation to favor this work's publication. I thank the authors for their kind response which not only answered my questions in a satisfactory manner, but also showed their disposition to address my concerns and potentially adopt some of my suggestions.

---

### Meta-Review · Area_Chair_NzJz · 2023-11-30

**Metareview:**

**Summary:** The paper introduces a novel mean-optical-flow feature called MDMO for micro-expression spotting.  It addresses the challenge of global movements in micro-expression analysis by presenting a skip-k-frame strategy and the M-pattern assumption.  The proposed spotting rules and MDMO features effectively remove disturbances from global movements while preserving micro-expression movements.  Experimental results demonstrate the superiority of the proposed methods compared to state-of-the-art approaches.

The reviewers agree that the paper has contributions and it should be accepted.  The reservations of reviewer KsQW are minor and the comments lean towards the acceptance of the paper.


**Strengths:** The paper effectively describes the limitations of the existing micro-expression spotting (MES) methods and proposes a method that addresses those limitations.  The use of MDMO and M-curve analysis is interesting and the experimental setup is well-designed to justify the effectiveness of the proposed method and design choices.  The elimination of global movements in the early feature extraction stage improves micro-expression detection.  The paper is well-written and the methodology appears technically sound.  The experimental results demonstrate that the proposed method outperforms state-of-the-art methods.  Overall, the paper presents an improvement over existing work in the field of ME spotting, utilizing pattern recognition and characteristics of actions effectively.


**Weaknesses:**  The reviewers identified issues with the presentation, including typos and a need for significant improvement in the paper's structure. It is also mentioned that the process of selecting hyper-parameters for the proposed method is not adequately explained. Additionally, there is a lack of theoretical analysis regarding why the M-patterns effectively mitigate the influence of global movements.

**Justification For Why Not Higher Score:**

The reviews acknowledge that the paper is technically correct, but they also note that the novelty and contributions of the paper are limited. However, there are no major flaws identified that would justify rejecting the paper.

**Justification For Why Not Lower Score:**

The reviewers have raised minor concerns, but they have not identified any major issues with the paper. Therefore, in my opinion, the paper should be accepted as it is technically correct.

---

### Decision · Program_Chairs · 2024-01-16

Accept (poster)